# The megabiota are disproportionately important for biosphere functioning

Brian J. Enquist [1,2]*, Andrew J. Abraham [3], Michael B.J. Harfoot[4], Yadvinder Malhi [5] & Christopher E. Doughty [3]

A prominent signal of the Anthropocene is the extinction and population reduction of the megabiota—the largest animals and plants on the planet. However, we lack a predictive framework for the sensitivity of megabiota during times of rapid global change and how they impact the functioning of ecosystems and the biosphere. Here, we extend metabolic scaling theory and use global simulation models to demonstrate that (i) megabiota are more prone to extinction due to human land use, hunting, and climate change; (ii) loss of megabiota has a negative impact on ecosystem metabolism and functioning; and (iii) their reduction has and will continue to significantly decrease biosphere functioning. Global simulations show that continued loss of large animals alone could lead to a 44%, 18% and 92% reduction in terrestrial heterotrophic biomass, metabolism, and fertility respectively. Our findings suggest that policies that emphasize the promotion of large trees and animals will have dispropor- tionate impact on biodiversity, ecosystem processes, and climate mitigation.

[1] Department of Ecology and Evolutionary Biology, University of Arizona, Arizona AZ 85721, USA. [2] The Santa Fe Institute, 1399 Hyde Park Rd, Santa Fe, NM 87501, USA. [3] School of Informatics, Computing, and Cyber Systems, Northern Arizona University, Flagstaff, AZ 86011, USA. [4] UN Environment Programme World Conservation Monitoring Centre, 219 Huntingdon Road, Cambridge CB3 0DL, UK. [5] Environmental Change Institute, School of Geography and the Environment, University of Oxford, Oxford OX1 3QY, UK. *email: benquist@email.arizona.edu

Perhaps the most conspicuous aspect of the diversity of life on the planet is the enormous range of the diversity of sizes of organisms. Indeed, large animals and trees (such as elephants, rhinos, whales, and large trees such as redwoods, sequoias, and mountain ash) are also often seen as charismatic and are often used as flagship species for conservation decisions. Large animals and trees are often highlighted as they have inspired much conservation effort and policy and effectively convey conservation principles to the public[1,2]. However, there is debate on if the charismatic nature of a species is a good indicator of conservation value or even a good predictor of conservation efficiency[1,3,4].

One of the primary signatures of the Anthropocene has been a progressive elimination of the largest organisms[5–8], especially if one of the first antecedents of the Anthropocene is argued to be the decimation of the Pleistocene megafauna[9]. Throughout most of the Phanerozoic, large animals and trees have been ubiquitous across the globe, except immediately following major extinction events in Earth history. Human activities are now disproportionately impacting the largest animals and trees[2,8]. This downsizing of the biosphere started in the Late Pleistocene with the extinction of much of the megafauna and continued through the rise of human societies marked by the exploitation of forests, ongoing hunting of large animals and clearing of land for agriculture and industry[10]. Here we coin the term megabiota to refer collectively to the largest plants and animals in the biosphere (i.e. the megafauna and megaflora). The megabiota are disproportionately impacted by land clearing, landscape fragmentation, hunting, overfishing, selective logging, human conflict, and climate change. As a result, populations of free ranging biodiverse megabiota on the planet have continued to be whittled down.

Since the rise of humanity it is primarily the largest and oldest trees that have become disproportionately rarer and more threatened. For example, since the start of civilization, the global number of trees on the planet has fallen by ~46%[11] and now ~15 billion trees are cut down per year[11]. However, it is the largest trees that have experienced the largest reductions. Primary and intact forests hold the largest trees, and these forest types are the most threatened (representing only one-third of all remaining forested land). Further, only 12–22% of primary and intact forests are largely safeguarded in protected areas, the remainder is vulnerable to exploitation[12]. Climate change and shifts in localized climate due to deforestation is now disproportionately impacting big and old trees. Large trees are declining in forests at all latitudes[2]. For example, in increasingly more fragmented rainforests half of the large trees (≥60 cm diameter) are at risk of loss just in the first three decades after isolation[13]. The density of the largest trees in Yosemite National Park have declined by 24% between the 1930s and 1990s[2,14]. Among the largest trees on earth, the mountain ash Eucalyptus trees (*E. regnans*) in Australia are predicted to decline from 5.1 in 1997 to 0.6 trees per hectare by 2070[2].

The average body mass of mammals on the continents has dropped precipitously with the spread of humans around the world[15]. Across the Earth today, it is the larger animals that are increasingly in peril, particularly predators[10,15–17]. With almost a quarter of large species at risk. Starting in the Pleistocene, due largely to hunting, large-bodied mammals have been systematically extirpated[17,18]. Importantly, the accelerated loss of large mammals also occurred during intervals that experienced combinations of regional environmental change including aridification and increased biome heterogeneity within continents[19] (Fig. 1). Today, of the world's largest carnivores (greater than or equal to 15 kg) and the world's largest herbivores (≥100 kg) 59% and 60% respectively are classified as threatened with extinction[20]. The major threats to the remaining megafauna include hunting, land-use change, and resource depression by livestock. Further, human conflict frequency (warfare etc.) predicts variation in population declines among wild large herbivores in protected areas in Africa, the last significant megafaunal continent, from 1946 to 2010[21].

Marine mammals have also seen broad population reductions due to widespread hunting over the past few hundred years[22]. Global fisheries have also been characterized by a reduction in the

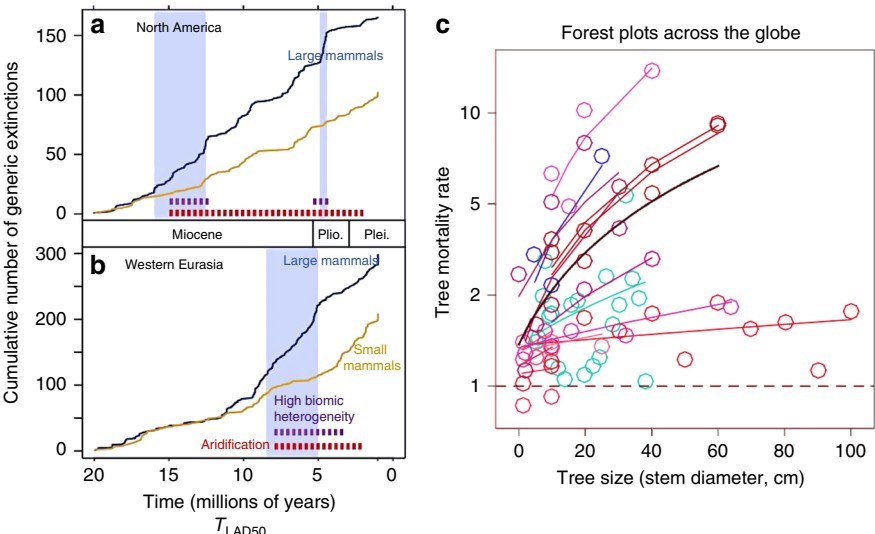

**Fig. 1 Larger body sized animals and plants are more susceptible to mortality and extinction in times of increased climatic stress.** The cumulative number of mammalian genus-level extinctions for large and small body size mammals plotted against the sampling-adjusted last appearance dates ($T_{LAD50}$). For animals, both North American (**a**) and western Eurasian (**b**) large (blue) and small (orange) mammals are shown. In both continents, phases of increasing drought (aridification; red broken bars) and fragmentation and heterogeneity of biomes is associated with elevated extinctions of larger mammals relative to those of smaller mammals (shaded areas). Graph modified from Tomiya[19]. For trees (**c**), larger trees exhibit greater increases in mortality rate relative to non-drought conditions. The different symbols and lines represent a unique drought instance within a given forest study (graph modified from Bennett et al.[39]). The dashed line is the expectation when tree mortality in non-drought conditions are similar to tree mortality in drought conditions.

mean and maximum size of fish in the ocean. Since the 1950s there has been a persistent and gradual transition from large long-lived, high trophic level, piscivorous fish toward smaller, short-lived, low trophic level invertebrates and planktivorous pelagic fish[23,24]. With climate change, the oceans will become warmer, more acidic, and contain less oxygen. Due to physiological requirements of fish, these changes are predicted to shrink the assemblage-averaged maximum body weight by 14–24% globally from 2000 to 2050 under a high greenhouse gas emissions scenario[25].

The reduction of the largest body sizes across of the diversity of life will increasingly have a major impact on the functioning of the biosphere[26]. However, given the scale of the problem, it is unclear if ecological theory can begin to predict the magnitude and extent of this perturbation on the biosphere[27]. We lack a general predictive framework to quantify how reductions in the size range of animals and plants will influence ecosystem and biosphere processes[28,29]. The rate of decline in the megabiota suggests that ever-larger regions of the world will soon lack many of the vital ecological services large organisms provide[30]. Therefore, there is an urgent need to understand how ecosystems change and may unravel with the decline of the megabiota[18,28,31].

Here we provide a theoretical underpinning to: (i) understanding why large animals and trees deserve conservation attention; (ii) the first set of comprehensive predictions for how the loss of the megabiota (the largest plants and animals) will impact (and has impacted) the biosphere; and (iii) policies that emphasize the promotion of large trees and animals on biodiversity, ecosystem processes, and climate mitigation. We first use analytical theory (Metabolic Scaling Theory or MST) to provide a foundation to generate a baseline set of predictions. We show that, in times of global change, the megabiota are more prone to extinction and decreases in their abundances disproportionately influence ecosystem and Earth system processes. We predict that compared to ecosystems where body size ranges are reduced ecosystems with megabiota will disproportionately house more biomass, carbon, and nutrients and will be more fertile. As a result, the loss of large organisms will decrease the metabolism and fertility of the biosphere. These findings also indicate that efforts to conserve both the megabiota and increase the area devoted to conserving the megabiota will have a multiplicative impact on ecosystem functioning. We assessed these predictions within a set global General Ecosystem Model (GEM) set of global simulations. We also test if potential variation in size scaling within complex ecological systems impact variation in ecosystem and biosphere metabolism. Our results underscore the importance of the megabiota to the functioning of the biosphere and to conservation priorities.

## Results

**Applying metabolic scaling theory to the megabiota.** Ultimately, cellular metabolism sets the pace of life and controls the flux of matter and energy in the biosphere[32]. The scaling of organismal metabolism powerfully constrains the functioning and life history of organisms across organisms from small to large sizes[33–35]. The scaling of metabolism sets the demand for resources, the space organisms require to forage, and the rate at which they interact with other organisms. Metabolism also influences the flux of energy and nutrients through organisms, populations, and ecosystems[33,36]. It constrains the rate of disease progression[37], the magnitude of how organisms interact with each other and their environment and influences their risk of extinction[18].

MST provides an analytical foundation to begin to understand the role of organismal size in ecology and evolution[32]. Building

on previous work, we derive a baseline set of predictions that show that the largest body sized plants and animals have a disproportionate impact on ecological systems. Our extensions of MST to the ecology and evolution of the megabiota (See Supplementary Information) makes five general sets of predictions:

**The megabiota have a higher risk of mortality and extinction.** The megabiota are more prone to population reductions and extinction than smaller body sized species due to the compounding effects of habitat loss, human hunting and harvesting, and climate change (Fig. 1). Future climate projections show that terrestrial regions will be characterized by hotter and more pronounced droughts, and oceans and freshwater habitats will be characterized by warmer temperatures, decreased pH, and reduced oxygen concentrations[25,38]. These factors will place additional physical limits on plant and animal size, and reduce available habitat. As a result, rapid sudden climate change will negatively impact the growth and survivorship of larger trees (Fig. 1c), fish, and aquatic invertebrates leading to reductions in body sizes and potentially exacerbating feedbacks to climate change[39] (Supplementary Information).

As we show in Supplementary Eqs. 4–5, Fig. 1a, b the probability of extinction, $E_\lambda$, in times of rapid climate change and/ or exploitation and habitat loss, will scale positively with body size[19]. This is due to three key characteristics of the megabiota. First, they often operate closer to biophysical, physiological, and abiotic limits. So, the risk of mortality due to extreme events, $R$, is more pronounced in times of rapid climate change[40]. Second, as they have lower per capita fecundity rates, $F$[41], their populations cannot rapidly rebound to change. Third, as a result, to maintain viable global population sizes, they require a larger minimum area, $A_m$ of habitat to avoid stochastic extinction[42]. Together, each of these characteristics scale with organism size, $m$, and combine to give a general allometric scaling expression for the probability of extinction, $E_\lambda$

$$E_\lambda \propto f\left[R(m^b) \cdot 1/F(m^{-c}) \cdot A_m(m^d)\right] \propto m^{b+c+d} \qquad (1)$$

We predict that during times of rapid habitat loss and climate change $E_\lambda$ will scale positively with body size (see also ref. [43]). Values of the scaling exponents are expected to approximate $b \sim 1$, $d \sim 1$, and $c \approx 0.25$ so that $E_\lambda \propto m^{2.5}$ (see Supplementary Information). Thus, as a rule of thumb, an organism that is 10 times larger in size will be about 316 times more susceptible to extinction during times of rapid change. Depending on the organism and environmental driver the values of $b$, $d$, and $c$ will likely vary indicating that we expect this rule of thumb to vary. Nevertheless, compared to smaller body sized flora and fauna, in times of rapid climate change and reductions of geographic range, larger body-sized species face disproportionally increased risk of extinction[19,44].

The findings of numerous recent studies are generally consistent with the above predictions. In times of rapid human land use and climate change, when compared to smaller flora and fauna, larger plants and animals face increased risk of mortality events[8,19,38,43,44]. Indeed, large trees are most susceptible to changing climate via warming temperatures and drought[39]. Compared to smaller trees, the biggest trees exhibit the greatest increases in mortality rate in hotter droughts relative to non-drought conditions[39]. An Amazon forest drought experiment has been simulating the impact of a moderate drought by reducing rainfall by a third in a 1-hectare forest plot[45]. In that experiment, tree mortality rates doubled for smaller trees but increased 4.5 times for the bigger canopy trees (Fig. 1). Similarly, the fossil record indicates that increasing drought and habitat

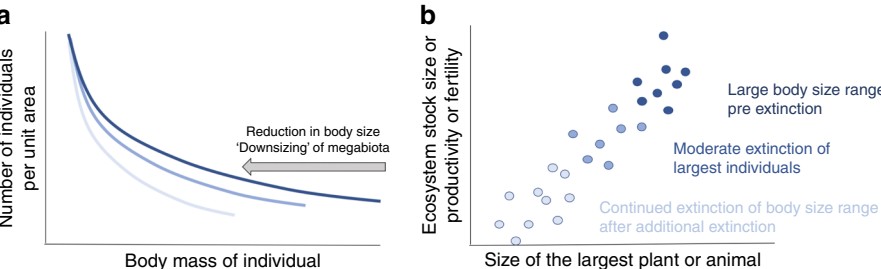

**Fig. 2 Conceptual diagram for how the downsizing of the biosphere (the sequential loss of the megabiota) influences the total amount of ecosystem stock (biomass, carbon, nutrients), productivity, or fertility.** In (**a**), for assemblages of either plants or animals, there is an inverse relationship between size and abundance. But, as larger organisms are disproportionately more prone to population reduction and extinction than smaller organisms, this leads to a reduction in the number of larger body sized individuals and a reduction in their numbers. As a result, past extinction and continued hunting, fishing, land and water use pressures in addition to climate change, are compressing body size distributions across most of the world's ecosystems. In (**b**) metabolic scaling theory and empirical data show that communities and ecosystems with larger body sized plants and animals flux more energy and resources. As a result, continued reductions in body size in (**a**) will lead to a continued reduction in ecosystem stocks and flux of energy and nutrients.

fragmentation are associated with elevated extinction rates of larger mammals relative to those of smaller mammals[19] (Fig. 1).

**The megabiota disproportionately impact ecosystem stocks and total biomass.** The megabiota disproportionately impact ecosystem functioning via influencing the scaling of total ecosystem standing stocks (e.g. the total amount of ecosystem carbon, nitrogen etc.; Fig. 2) and biomass (Supplementary Eqs. 6–9). This impact is the result of two important ecological factors—the size spectra (the distribution of the sizes of all plant or animal individuals found in a given location, Fig. 2a), and, $f$, the allometric relationships that characterize how structural attributes and physiological/metabolic rates of an individual change or scale with differences in body size. Depending on the environment, plants and animals can fill and occupy space differently (three-dimensional packing of roots and canopies vs. more two-dimensional packing of animal home ranges and territories). As a result, the impacts of the megabiota can differ depending on their ecology.

In the case of terrestrial plants (autotrophs), the total biomass of the forest, $M_{Tot}$ can be related by a primary size measure—the radius of the plant stem, $r$, and the size distribution of the stems in that forest, $f(r)$ where $f(r) = cr^{-\eta}$ (see Supplementary Eq. 7). The value of the exponent, $\eta$, may vary but is hypothesized to approximate $\eta \approx -2$ in undisturbed forests, a value supported by empirical studies[46]. Using idealized allometries, the total phytomass of an individual, $m$, can be related to the primary size measure—stem radius of a tree, $r$, where $m(r) = c_m^{8/3}r^{8/3}$, where $c_m$ is an allometric constant that may vary within or across taxa. We can then derive a general scaling law relating $M_{Tot}$ and the size of the largest plant's stem radius, $r_{max}$,

$$M_{\text{Tot}} = \int m(r)f(r)\mathrm{d}r = \int \left(\frac{r}{c_m}\right)^{\frac{8}{3}}(c_n r^{-2})\mathrm{d}r$$
$$\approx \left(\frac{3}{5}\frac{c_n}{c_m^{8/3}}\right)r_{\max}^{5/3} \tag{2}$$

As the trunk radius of the largest tree in the forest increases, the total forest biomass, $M_{tot}$, increases disproportionately faster. Specifically, total biomass increases as the size of the largest individual tree raised to the 5/3 or 1.67 power of its trunk radius, $r_{max}$. Expressed as a function of the mass of the largest tree in the forest, $m_{max}$ (kg), the total forest biomass increases as the $M_{Tot} \propto m_{max}^{5/8}$ (Supplementary Information). So, the total amount of biomass contained within the forest increases as the 5/8 or 0.625 power of the mass of the largest tree in the forest.

Similarly, in the case of animals (applied to all individuals within a trophic level), the total biomass of a trophic group, $M_{Tot}$ can be related by its primary size measure—organism biomass, $m$. The size frequency distribution of all animals is measured in terms of animal mass, $f(m)$ where $f(m) = cm^{-\epsilon}$. The value of $\epsilon$ may vary but is hypothesized to approximate $\epsilon = -3/4$[47]. The total biomass of all animals in that trophic level, $M_{Tot}$ is predicted to scale with the size of the largest animal, $m_{max}$ (see Supplementary Information) as

$$M_{\text{Tot}} = \int m(m)f(m)\mathrm{d}m = \int m \cdot c_a m^{-3/4}\mathrm{d}m$$
$$\approx \frac{4}{5}c_n m_{\max}^{5/4} \tag{3}$$

This predicted relationship, indicates that, in a given trophic level, as the mass of the largest animal increases, the total trophic biomass of all animals increases disproportionately faster. When expressed in terms of organismal biomass, this predicted superlinear scaling of total trophic biomass, shows that changes in maximum size of an animal $m_{max}$ will have a larger and disproportionate impact on the total trophic biomass $M_{Tot}$ (see Supplementary Fig. 1)

We tested these predictions via several different approaches. Observations of forests across the globe, in both temperate and tropical forest communities (Fig. 3) show that the size of the largest individual, $m_{max}$ is a strong predictor of total forest biomass, $M_{Tot}$. The fitted scaling exponent for total forest biomass, 0.62 (95% CI = 0.58–0.66), is indistinguishable from the MST prediction of 5/8 = 0.625 (see ref. [48]; Fig. 3). As we discuss below, global simulation models that incorporate metabolic and allometric scaling also show the predicted positive scaling relationship between body mass and total heterotrophic biomass (see below), but the relationship is modified by local climate.

**The megabiota disproportionately impact ecosystem flows.** MST predicts that the megabiota impact ecosystem functioning via their disproportionate impact on total trophic biomass which then drives the total metabolic and resource fluxes and ecosystem net primary productivity[33]. For autotrophs, the total energy flux through all plants, $B_{Tot}$ and the total net biomass productivity or net primary productivity or NPP (or the total resource flux $J_{Tot}$) scales with the size of the largest individual and the total autotrophic biomass. In Supplementary Eqs. 10–13 we derive a general scaling law for how total trophic biomass, $M_{Tot}$, influences variation in ecosystem fluxes including total energy, $B_{Tot}$, biomass productivity, NPP, and carbon, and nutrients. The total resource

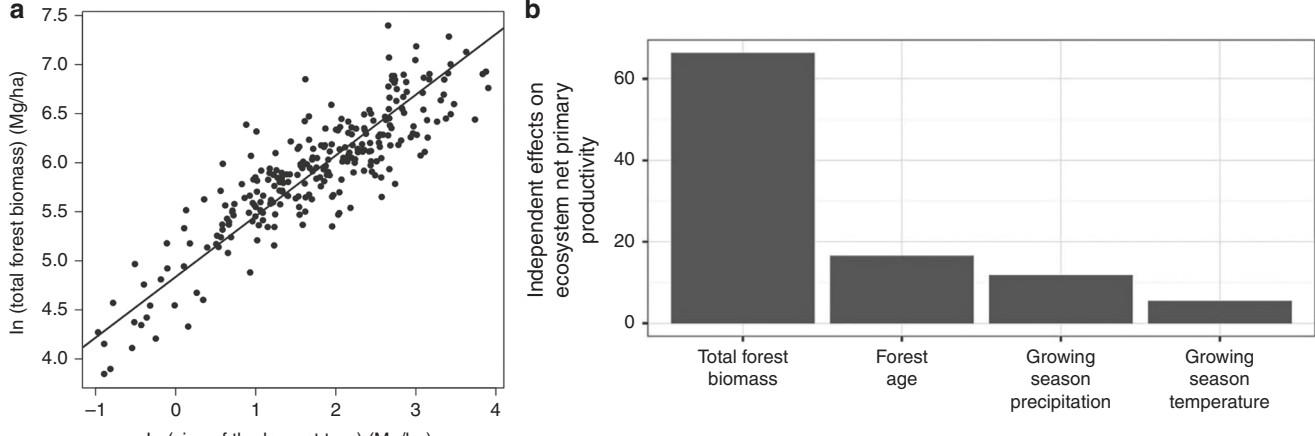

**Fig. 3 Forests with larger trees disproportionately store more biomass (carbon) and are more productive.** In (**a**) the total above ground forest biomass is best predicted by the size of the largest tree. Analysis of biomass calculated from $n = 267$ independent forest plots distributed across the Americas from 40.7° S to 54.6° N latitude. The best single predictor of variation in forest biomass is the size of the largest tree in that forest. The fitted slope of the relationship (the scaling exponent) is 0.62, which is indistinguishable from the predicted scaling function from metabolic scaling theory where the total biomass should scale as maximum tree size to the 5/8 or 0.625 power. Data from ref. [48]. In (**b**) global analyses of the relative importance of several drivers of variation in forest ecosystem net primary productivity (data from ref. [51]). The most important driver of variation in terrestrial net primary productivity is the total forest biomass. Variation in forest biomass has a larger effect than precipitation, temperature, and forest age. As the best predictor of total forest biomass is the size of the largest individual (**a**) these results indicate that forests with large megaflora are more productive. Vegetation with megaflora collectively dominate the biomass and carbon stored in vegetation and the productivity of land vegetation.

utilization rate $J_{Tot}$ (kg yr$^{-1}$) of a given resource $i$, such as nitrogen, water, carbon etc, can be written as

$$J_{Tot} \propto NPP \propto B_{Tot} \approx \left( \tau \kappa_i^{-1} B_0 c_n \right) r_{max} \qquad (4)$$

As the size of the largest tree within a forest increases, the total system flux will scale in direct proportion to the largest individual. The total amount of resources (carbon, water, nutrients) that pass through the ecosystem or through a food web will increase as maximum tree height increases. In terms of the total autotrophic biomass, as the size of the largest tree influences total forest biomass, $M_{Tot}$, (Eq. 2) and NPP, we can relate NPP to $M_{Tot}$ as NPP $\propto B_{Tot} \approx b_0 c_m^{8/5} c_n^{2/5} [5/3 M_{Tot}]^{3/5}$. Thus, forests with trees 10 times larger in trunk diameter will store ~47 times more carbon (see above, Eq. 3) and will assimilate 10 times more carbon and produce 10 times more biomass. As a result, vegetation that contains larger individuals will disproportionately absorb and store more carbon and cycle more water and nutrients and in turn produce more biomass.

Similarly, for animals, because of the allometry of resource use and packing of ecological space, we have a similar but slightly different scaling relationship indicating that increases in the maximum body mass of an animal would also disproportionately increase the total amount of flux through the heterotrophic food web. With substitution, we then have

$$J_{Tot} \approx \left( \tau \kappa_i^{-1} B_0 c_n \frac{4}{5} c \right) m_{max}^{5/4} \qquad (5)$$

Importantly, for animals, the flux of energy and matter through the heterotrophic food web is predicted to scale to the 5/4$^{th}$ or 1.25 power of the total heterotrophic biomass. Ecosystems with the largest animals 10 times larger in mass flux ~18 times more energy and nutrients (see above, Eq. 3). Thus, as the size of the largest individual (as measured by the primary size) within a given trophic group increases, the *total* ecosystem trophic flux will scale superlinearly.

Support for the above MST predictions are shown in Fig. 3, Supplementary Information, and by recent studies assessing the dynamical predictions for ecosystems[46,49,50]. Variation in forest

biomass has a larger effect on variation in ecosystem productivity (NPP) than precipitation, temperature, and forest age[51]. Similarly, the best predictor of forest biomass is the size of the largest individual (Fig. 3a), together these results show that forests with large megaflora are more productive and contain more stored carbon (Fig. 3). For animals, tentative support for this prediction is given by earlier macroecological analyses where species of large body sized birds flux more energy than small body sized birds[52].

**The megabiota disproportionately impact ecosystem fertility.** Larger herbivorous animals are disproportionately more important in the lateral movement of nutrients and energy in the biosphere via dung, urine and flesh. This movement takes two main forms: diffusion and directional transport. Recent work has utilized aspects of metabolic scaling theory to quantify the movement of nutrients across space by herbivores[53]. We show that MST makes specific predictions for the scaling of nutrient diffusivity in ecosystems as a function of the largest sized animal ("Methods"; Supplementary Information). Specifically, the diffusion of nutrients across the landscape by herbivores via defecation and urination, $\phi$, scales positively with the size of the largest herbivore, $m_{Herbivore}$.

$$\phi \propto m_{Herbivore}^{1.17} \qquad (6)$$

We assessed these predictions, by (i) simulating how a reduction in body size of herbivores in Amazonian forests affects the distribution of soil phosphorus across the Amazon basin (see the "Methods" section); and (ii) implementing the allometric scaling of metabolism and animal movement in a global simulation model (see below). Consistent with predictions, the Amazonian simulations show that the observed reduction in the size range of the megafauna in the Amazon from the Pleistocene baseline leads to reduction in ecosystem fertility as measured by steady state soil phosphorus concentrations (Fig. 4). Under a series of size thresholds for the extinct megafauna, we expect a 20–40% reduction in soil steady state P concentrations. Recent empirical studies are consistent with these predictions and point to the

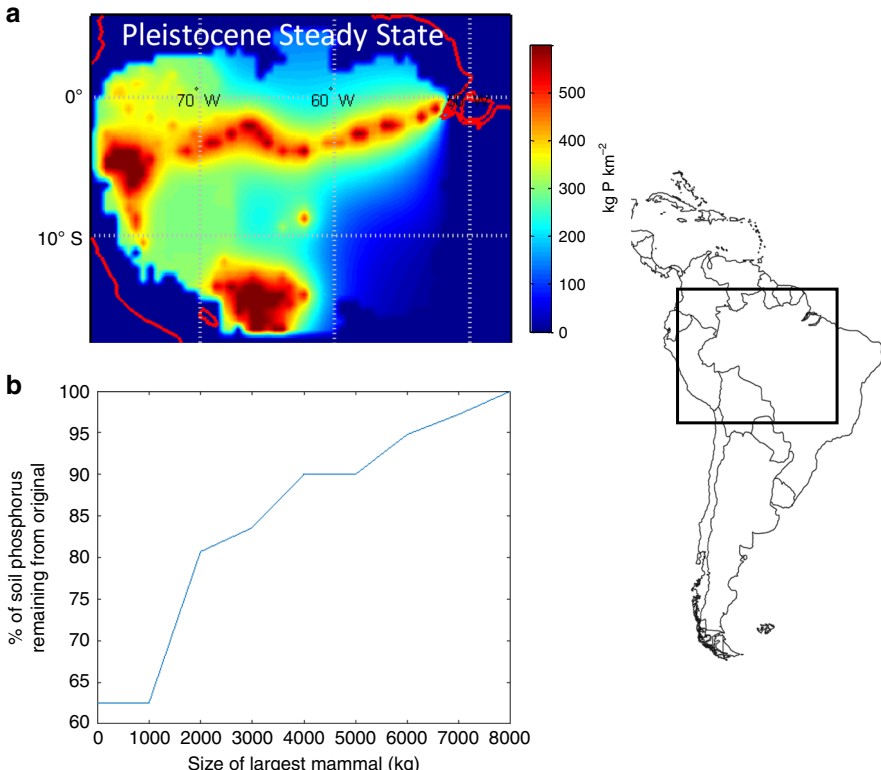

**Fig. 4 A decrease in the mean size of South American fauna is predicted to decrease soil fertility (phosphorus) throughout the Amazon basin.**
Predictions for how the steady state fertility of the Amazon basin (soil phosphorus concentrations) has changed in response to the megafaunal extinctions. This simulation is characterized by lateral diffusivity of nutrients, $\Phi$, by mammals away from the Amazon river floodplain source. The diffusivity of nutrients through the Amazon via ingestion, transport, and eventual defecation yields a $\Phi$ value of 4.4 km$^2$ yr$^{-1}$ (based on Doughty et al.[27]). **a** Simulated steady state soil P concentrations across the Amazon Basin with the now extinct megafauna; **b** With the extinction of large mammals and a continued forecasted reduction in mammal body size, the percentage of original steady state P concentrations in the Amazon Basin will decrease. Here, under a series of size thresholds for the extinct megafauna, we expect a 20–40% reduction in soil steady state P concentrations. For instance, a 5000 kg size threshold removes all animals above 5000 kg and continental P concentrations are reduced by ~10%. A size threshold of 0 has all extant South American mammals. Amazonian map from MATLAB worldmap from the Global Optimization Toolbox, The MathWorks, Inc. www.mathworks.com.

importance of megafauna on nutrient redistribution and fertilization of ecosystems[54,55].

**Conservation implications of the multiplicative importance of the megabiota and total area protected.** The megabiota are also disproportionately more impactful for conservation efforts prioritizing ecosystem functioning. For example, because the total biomass of a given trophic level, $M_{tot}$, will be directly proportional to the amount of area $A$ ($M_{tot} \sim A$) protected[56], doubling the area available for the megabiota will further have a disproportionate effect on ecosystem functioning (see Supplementary Eq. 16; Supplementary Fig. 1B). For example, efforts to maximize ecosystem services can be amplified by effort to maintain and increase large body sized plants and animals (Supplementary Fig. 1) and also conserve larger areas. Allowing for increases in maximum organism size and allowing more area to be restored to forest or to rewild large animals[57] will together have a multiplicative and nonlinear effect on ecosystem services (Supplementary Fig. 2).

**Global simulations of the biosphere with and without megaherbivores.** One of the limitations of the above derivations from MST is that the analytical theory does not yet tackle the complexity of species interactions in differing landscapes. In particular, including how ecological interactions and dynamics within and across differing landscapes and trophic levels have yet to be detailed by MST. Removing the megabiota does more than just reduce the body size range of plants and animals—it changes how individuals and species interact with each other[58]. These networks of ecological interactions are also fundamentally altered by shifting the relative importance of competitive and mutualistic interactions and the presence of trophic cascades[10]. For example, loss of the megabiota could influence the growth and abundance of smaller plants and animals. Their response could then compensate for ecosystem functions and possibly negate the above predictions. To more fully assess how downsizing of the planet's fauna will influence ecosystem processes within the context of complex species interaction networks we utilized a General Ecosystem Model (GEM).

We used the Madingley Model as it explicitly incorporates the importance of organismal body size (metabolic demands, foraging area, and population dynamics[59]; see Supplementary Fig. 2). This formulation of a GEM represents complex ecological interaction networks and whole-ecosystem dynamics at a global scale[60]. It is capable of modelling emergent ecosystem and biosphere structure and function by simulating a core set of biological and ecological processes for all terrestrial and marine organisms between 10 μg and 150,000 kg. Details of the simulation model are described in the methods section, Supplementary Information, supplementary Figs. 2–9).

We generated a set of forecasts for how, since the Pleistocene, the downsizing of the terrestrial megafauna has altered or will alter the functioning of ecosystems and biosphere. We ran three sets of simulations, or three different worlds (see "Methods" and

**Table 1 The global total of the three-heterotrophic ecosystem-level measures.**

| | Pleistocene world | Modern world | %Reduction Pleistocene to Modern | Future world | %Reduction Modern to Future | %Total change from Pleistocene to Future |
|---|---|---|---|---|---|---|
| Global Heterotroph Biomass (Pg) | 23.60 (23.13, 24.10) | 18.00 (17.74, 18.13) | −23.7% (−21.6, −26.4) | 13.20 (12.85, 13.55) | −26.7% (−25.3, −29.1) | −44.1% (−43.8, −46.7) |
| Global Heterotroph Metabolism (EJ/day) | 4.04 (4.01, 4.07) | 3.80 (3.76, 3.83) | −5.9% (−4.5, −7.6) | 3.32 (3.29, 3.38) | −12.6% (−11.8, −14.1) | −17.8% (−17.0, −19.2) |
| Global Nutrient Diffusivity Potential ($10^7$ km$^2$/day) | 3.01 (2.90, 3.12) | 0.80 (0.78, 0.83) | −73.4% (−71.4, −75.0) | 0.23 (0.22, 0.24) | −71.3% (−71.1, −73.5) | −92.4% (−92.3, −93.0) |

Results are derived from the ensemble General Ecosystem Model (GEM) simulations of the Pleistocene world, Modern world, and Future world. Values in parentheses are 95% confidence intervals. Percentages compare the difference between the Modern and Future worlds to the Pleistocene world. The percent total reduction compares how each of these global ecosystem functions are predicted to decrease from the baseline Pleistocene biosphere with a full component of large animals to a future world lacking the animal megabiota.

Supplementary Information). In each world, we simulated the loss of the endotherm herbivore megafauna by experimentally changing the maximum attainable body mass. Each world differed in maximum size by an order of magnitude, from 10,000 kg (the largest terrestrial Pleistocene herbivore, *Mammuthus columbi*), to 1000 kg (typical modern day maximum size of terrestrial mammalian taxa) and finally 100 kg (a future world lacking wild megaherbivores). The body mass ranges for all other terrestrial animal cohorts were held constant and approximating those found in the Pleistocene fossil record (see ref. [60]; Table 1; see Supplementary Fig. 2). We hereafter refer to these three worlds as (i) Pleistocene world, (ii) Modern world; and (iii) Future world.

Multiple lines of evidence from the GEM simulations (Fig. 5a–c) are consistent with predictions from MST (Eqs. 2–6; see also Supplementary Eqs. 9, 11, 13–15). We observed a disproportionate impact of the megabiota with a positive, but increasing, relationship with maximum body size and ecosystem function (Fig. 5d–f),. Reductions in the size of the largest animal—megaherbivores—leads to a decrease in biosphere functioning (Table 1; Fig. 5d–f). Compared to the Pleistocene baseline, the total biosphere biomass is predicted to decrease 44.1% from 23.60 Pg to 13.20 Pg (Fig. 5d, Table 1). The impacts of megaherbivore loss vary spatially indicating that local climate and species composition may further modify MST predictions. Large impacts of megaherbivore loss are observed in sub-tropical regions of the world (see Supplementary Figs. 4,8–9) because these regions are characterized by the largest animals (see Supplementary Fig. 4c, e). Reductions in maximum herbivore body size have the greatest impact on ecosystem nutrient diffusivity, with global measures of future nutrient diffusivity decreasing by 92.4% between the Pleistocene and Future worlds (Fig. 5f, Table 1). The loss of megaherbivores in a future world has a smaller impact on global heterotrophic metabolism (decreasing 18%; see Table 1).

We also tested an important alternative hypothesis—with the loss of the megabiota, the response of smaller organisms could compensate for the loss of the megabiota. Specifically, with the loss of ecological interactions from the megabiota, ecological and evolutionary responses from smaller organisms could lead to changes in their abundance and range[61] and compensate for the loss of large herbivores and carnivores. We used the GEM to test if smaller animals experience an ecological release with the loss of the larger body plants and animals, and if they can they provide the ecosystem functions of the megabiota. Our results indicate that while there is some compensation from the smaller organisms in terms of heterotrophic metabolism (Fig. 5e), we see little to no compensation in global heterotrophic biomass and nutrient diffusivity (Fig. 5d, f).

## Discussion
There has been considerable debate, on whether conservation goals are best achieved by promoting management of a single charismatic species or focusing on whole-ecosystem functioning[1]. Charismatic species in conservation are most often large mammals and vertebrates[1,62], although large old growth trees and old growth forests can also be charismatic[63]. Conservationists have argued that actions intended to preserve an iconic charismatic species can have an umbrella effect and save less-glamorous species and whole ecosystems that thrive in its shadow. However, can large organisms act as a proxy for the diversity and functioning of whole ecosystems[31]? Such proxies are difficult to measure. The natural charisma of large animals and trees is often cited as the best justification to protect habitat and entire ecosystems[64]. Nonetheless, considerable debate remains. Daniel Simberloff, noted that "whether many other species will really fall under the umbrella is a matter of faith rather than research"[4]. A worry is that while only charismatic species seem able to appeal enough interest to raise sufficient funds and interest a focus on the large charismatic fauna and flora is not based on science[65].

Our simulation results are consistent with the arguments of Estes et al.[10] who underscored that ecological theory based on species trophic interactions implies that downsizing of the biosphere will result in major shifts in ecosystem and biosphere functioning (Supplementary Information). Ecological theory based on species interactions further points to the importance of the megabiota in also influencing other aspects of ecosystem functioning tied to human health and well-being. For example, there are strong lines of evidence to suggest that loss of the megabiota negatively impacts ecosystem resilience to climate change, human health via disease dynamics (influencing emerging diseases and pathogenesis), biological diversity, and buffering ecosystem functioning[66]. We are only starting to understand the connections between human health and the megabiota but preliminary data and extensions of MST to pathogenesis and ecosystem resilience points to important linkages[37].

Together, our theory and simulations indicate that many conservation and climate change mitigation policies can be assisted by emphasizing the conservation reestablishment and promotion of the largest organisms. The widespread extinction of megafauna and decline in abundance of many remaining megafauna have progressively eliminated an interlinked biosphere system for the recycling of nutrients and reducing the metabolism of the biosphere. In a world with megabiota more carbon and nutrients are stored in vegetation and through animal movements, they flow against entropy from the ocean depths to continental interiors and from fertile soils to relatively poorer soils[67]. Our results support past speculations that a reduction in the largest animals will result in a drop in nutrient diffusion capacity[67]. A decrease in nutrient concentrations in regions that are distant from their abiotic sources result in broad global regions being less fertile[27,67]. Simply put, landscapes and ecosystems that contains larger and more abundant organisms are more productive, more resilient to climate change, and will provide disproportionately more ecosystem services to humanity.

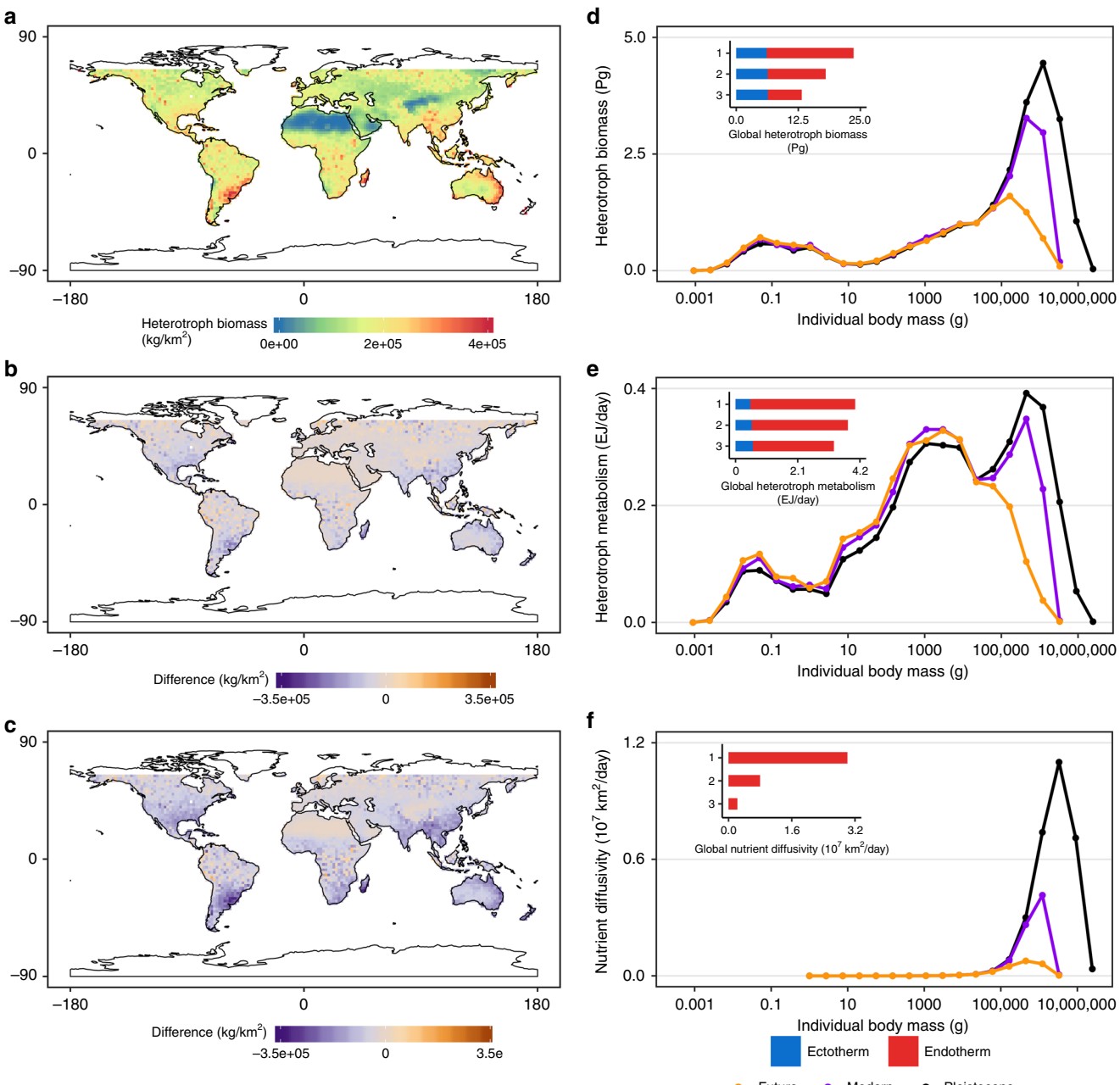

**Fig. 5 The continued global downsizing of large animals results in reductions in terrestrial biosphere biomass, metabolism, and fertility.** The annual mean heterotrophic community biomass from three ensemble experiments using the General Ecosystem Model (GEM) mapped spatially showing **a** the Pleistocene world, **b** the difference between the Pleistocene world and Modern world and **c** Future world. The implications of this downsizing for the functioning of the biosphere are measured using the annual mean of the GEM experiments for the three ecosystem-level measures; **d** heterotrophic biomass, **e** heterotrophic metabolism and **f** nutrient diffusivity summarized into 25 mass bins. The inset graphs display the global total for each metric and are numbered (1) Pleistocene, (2) Modern and (3) Future world respectively. Global map from the 110 m land polygon shapefile from Natural Earth Data (https://www.naturalearthdata.com/downloads/110m-physical-vectors/).

Our growing understanding of the role of the megabiota raises numerous questions that future research will need to address. In Supplementary Box 1 we detail a set of questions that stem from our findings (see Supplementary Box 1; Supplementary Information). Key questions remain in terms of how reductions of the largest sized individuals influence complex ecological networks, human health, and biosphere functioning. For example, in a world with fewer megabiota, how will the dynamics of ecosystems and biodiversity change[7,18]? Because large organisms are long lived and their population cycles are longer, the presence of large-bodied organisms can buffer ecological systems from environmental variation. Will ecological systems and human interactions with ecological systems (fisheries, forestry) become less buffered with time with loss of the megabiota? How long will it take reforestation and restoration efforts to revive ecosystem processes promoted by large body sized species[68]?

It is also important to emphasize that while the wild megabiota has greatly decreased, that nature of the megabiota has profoundly changed and become domesticated[18]. Our results do not incorporate increases in the population of large domestic animals (domesticates such as cattle, pigs etc.) and forest plantations and management that have greatly increased. Further, an important

question is to what extent the anthropocentric megabiota functionally compensate for or exacerbates the impacts of lost wild animal biomass. Some forms of megabiota domestication and management may replicate the functioning of wild megabiota e.g. nomadic pastoralism, forest and woodland management. Many other forms, such as industrialized animal farming and forest management with restricted animal movement and limits on tree size from fencing and human land use, do not.

Our results do not argue or indicate that smaller organisms are less important or that they should be ignored. Indeed the role of the smaller organisms (e.g. microbes[69], insects[70], etc.) are crucial to ecosystem and biosphere functioning. Our point is that the functioning of the biosphere and the well-being of increasingly smaller organisms disproportionately relies on the largest organisms. Further, the smaller organisms cannot provide most of the distinctive ecological roles and services played by large old trees and animals. Nonetheless, more research is needed to help understand uncertainties and clarify limits of our theoretical predictions numerous lines of evidence point to the disproportionate impact of the megabiota to the functioning of the biosphere.

In conclusion, we have presented a theoretical framework and a global simulation model that provides a set of baseline predictions for how the loss of the megabiota will influence several aspects of ecosystem structure and function and tested several predictions of metabolic scaling theory. Our analytical theory predicts that forests and animal communities with larger body sizes will disproportionately contain more biomass, carbon, and nutrients and disperse nutrients further than ecosystems where body size ranges are reduced. Further, as the land area devoted to conservation of megabiota increases, the megabiota have a multiplicative impact on total biomass and ecosystem functioning.

There is an urgent need for interdisciplinary research to forecast the effects of trophic downsizing on process, function, and resilience within ecosystems and the biosphere[10] (see Supplementary Box 2). In this paper, we introduced the term megabiota to refer to the biosphere consisting of the largest plants and animals. We provided an overview and extended metabolic scaling theory to show how MST can be used to provide a set of strong predictions for the importance of the largest plants and animals for ecosystem structure and functioning. We utilized a global simulation model to more fully assess and explore several predictions from metabolic scaling theory. Both theory and our simulation results show the disproportionate importance of the megabiota on the impact on ecosystems and the functioning of the biosphere.

Our results show that a biosphere with larger plants and animals is more productive, contains more biomass, and is more fertile than a biosphere lacking in the largest animals. Further, it is also increasingly clear that a biosphere with megabiota is more buffered, resilient, and positively influences biological diversity. There is mounting evidence that the megabiota, via how they influence ecological interactions, encapsulate the checks and balances that minimize boom-and-bust cycles of species outbreaks, disease dynamics, and ecosystem imbalances (Supplementary Information). The result is a benefit to human health and economies by minimizing biological ecosystem stochastic variation and increasing the fertility and productivity of the biosphere. Ecological systems that are missing these key regulatory players, such as large predators, herbivores, and trees, provide fewer ecosystems services, are less predictable, and can collapse[10]. While there are important caveats and uncertainties (Supplementary Box 2) promoting the conservation and management of the largest organisms enhances numerous linkages to whole-ecosystem diversity, functioning, and services. The continued reduction of the megabiota will have long lasting and profound impacts on the Earth System that are not included in our current earth system models[18]. We are only starting to realize and quantify these impacts. Conservation and climate mitigation policies that emphasize the conservation, reestablishment, and promotion of the largest trees and animals will have more impact on biodiversity and ecosystem processes than polices that do not prioritize the megabiota.

## Methods

**Analytical derivations.** Detailed analytical derivations are given in the Supplementary Information in Supplementary Eqs. 1–16.

**Amazon fertility simulation.** Following Doughty et al.[27], we modelled how reductions in the maximum size of herbivores would then impact the fertility of the soils of the Amazon Basin[27]. We calculated the steady state estimate of soil P concentrations in the Amazon basin prior to the megafaunal extinctions. The extinctions of the megafauna in South America has led to drastic changes in animal size distributions with 70% of mammal species greater than 10 kg going extinct (62 species) since the Pleistocene including such large iconic species as gomphotheres, giant sloths and glyptodonts. This simulation is characterized by lateral diffusivity of nutrients ($\Phi$) by mammals away from the Amazon river floodplain source. In the simulation, animal nutrient transport are modelled via diffusivity of nutrients via ingestion, transport, and eventual defecation.

**GEM simulations.** The conceptual basis for GEMs is described by Purves et al.[60] and a comprehensive explanation of the GEM are provided by Harfoot et al.[59] The Madingley model GEM explicitly simulates the dynamics of plants and all heterotroph organisms between 10 µg and 150,000 kg. The model is mechanistic, generating emergent ecosystem structure and function by simulating a core set of biological and ecological processes at the level of an individual. The GEM is global in scope, but is spatially and temporally flexible, allowing for application at regional and local scales and can be applied in both the terrestrial and marine realms.

On land, plants are represented by stocks of biomass simulated using the climate-driven model of Smith et al.[71]. Plant biomass is added to the autotroph stock each time step in each grid cell through environmentally-driven primary production, the seasonality of which is calculated using remotely-sensed Net Primary Productivity or NPP. This production is allocated to above-ground/below-ground, structural-non-structural and evergreen/deciduous components as a function of the environment. Biomass is lost from plant stocks through mortality from fire, senescence and herbivory.

Our simulations do not directly change the available plant biomass or abundance of animal cohorts modelled within each grid cell, which is instead a function of environmental suitability and ecological pressures. Nor do these simulations include future land use, historic land use, the rise of domesticated animals (cattle, pigs etc.) or differing climate change scenarios. Instead, to start, we assess the effects of reducing the maximum size of just warm-blooded herbivorous animals. Due to stochasticity generated within the GEM, we performed an ensemble of five 100-year global simulations for each world using a monthly time step and a resolution of 2° × 2° grid cells. To understand the importance of the loss of megabiota in one trophic level (megaherbivores) in shaping ecosystems, we considered three ecosystem-level measures; total heterotrophic biomass, total heterotrophic metabolism and nutrient diffusivity (for further details on the calculation of these metrics please refer to Supplementary Information).

**GEM experimental constraints.** The complex network of dynamic ecological interactions modelled within the Madingley GEM model can lead to unpredictable behaviour. Consequently, to constrain our analysis to a single cause we circumscribed our experiments to manipulating the body size of endothermic herbivores only. For this, we exclusively modelled the terrestrial realm because the megafauna extinctions have been less severe in the oceans than on land.

The constraining characteristics of cohorts used in the three experiments are summarized in Supplementary Table 1. For all cohorts except endothermic herbivores, these properties were kept as those used by Harfoot et al.[59], which are broadly realistic of the late Pleistocene prior to the megafauna extinctions[73]. We use the same annual time-series of monthly climatological input data for each year of the 100-year simulation, and importantly do not include the effects of anthropogenic habitat conversion or the harvesting of plants and animals. As a result, except for climatic conditions which are based on averaged data between 1961 and 2000, our simulations represent a late-Pleistocene ecological world.

We experimentally changed the maximum attainable body size of endothermic herbivores across two orders of magnitude from 10,000 kg to 100 kg. This upper value is representative of the largest known herbivorous mammal, *Mammuthus columbi*, as estimated from the fossil record by Faurby and Svenning[73]. Importantly, these upper and lower body masses determine the possible range over which cohorts can theoretically be realised in each grid cell. Once the model is running, however, environmental and ecological pressures may not allow these limits to be reached.

**Heterotroph biomass.** The GEM is not deterministic, so we performed an ensemble of five 100-year global simulations for each world using a monthly time step and a resolution of 2° × 2° grid cells. After 100 model years simulations reached a dynamic steady state. Heterotrophic organisms were assigned to one of 25 mass bins based upon the natural logarithm of their individual body mass. To capture the seasonal variations in grid cell heterotroph biomass, we calculated the mean heterotrophic biomass in each grid cell using the last 12 monthly time steps, or one whole year. This was performed for the five simulations run for each world, from which an ensemble mean value was derived for each terrestrial grid cell. Heterotroph biomass within each grid cell and globally was then summarised by trophic group, thermoregulatory strategy and mass bin for use in our experimental analysis.

**Heterotroph metabolism.** The metabolic rate, in kJ/day, for each organism was calculated assuming a power-law relationship with body mass and an exponential relationship with temperature following[74]. Whilst an organism is active, metabolism is described by field metabolic rates, whilst when inactive metabolism is described by basal metabolic rates (see Supplementary Eq. 2 and ref. [59]). Endothermic functional groups were considered to be metabolically active for the entirety of a time step with a body temperature of 310 K. For ectothermic functional groups, the proportion of time active and body temperature was derived from the ambient temperature $T^K$.

**Nutrient diffusivity.** Nutrient diffusivity was calculated from a modified version of Eq. 3 in Wolf et al.[53]: Next, because our GEM can calculate animal density, we then calculated $\Phi_{excreta}$ by inputting animal density predicted by our GEM into Supplementary Eq. 3.

**Reporting summary.** Further information on research design is available in the Nature Research Reporting Summary linked to this article.

## Data availability
The datasets generated during and/or analyzed during the current study are available in the via our github repository https://github.com/andrewjabraham/Megabiota.

## Code availability
The source code underlying the Amazon phosphorus simulation (Fig. 4) in Matlab is available via GitHub https://github.com/andrewjabraham/Megabiota. The C# code to run the GEM Madingley simulation is available via GitHub (https://github.com/Madingley/C-sharp-version-of-Madingley). There are good instructions here on how to run the model here should any readers wish to do so. The R codes to analyse the GEM simulations and recreate all associated figures and tables (and Fig. 5, Supplementary Figs. 2-8 as well as Table 1 and Supplementary Table 2) are available via GitHub https://github.com/andrewjabraham/Megabiota.

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

## Acknowledgements

B.J.E., Y.M., M.B.J.H. and C.E.D. thank the Gordon Research Conference on Metabolic Ecology for providing the atmosphere to help generate the early seeds of the ideas that spurred this work. We thank members of the Enquist Lab for feedback on early discussion of these ideas. C.E.D. and A.J.A. acknowledge funding by NASA award 16-HW16_2-0025 and a Google Earth Engine research award, Y.M. by the Jackson Foundation. M.B.J.H. by a KR Foundation grant FP-1503-01714, and B.J.E. by NSF award DEB 1457812. B.J.E. was also supported by a Visiting Professorship Grant from the Leverhulme Trust, UK and an Oxford Martin School Fellowship.

## Author contributions

B.J.E, A.J.A, Y.M., M.B.J.H. and C.E.D. designed the research; B.J.E. and A.J.A. gathered data; A.J.A., M.B.J.H., and C.E.D. carried out simulations; A.J.A. and B.J.E. analyzed the data; B.J.E. and Y.M. carried out the analytical derivations; B.J.E. wrote the first draft of the manuscript; B.J.E., A.J.A., Y.M., M.B.J.H. and C.E.D. contributed to the revisions.

## Competing interests

The authors declare no competing interests.
