## [Peer Review File · Nature Communications]

Reviewers' comments:

Reviewer #2 (Remarks to the Author):

I have reviewed this manuscript in the previous round (Referee 5). I appreciate the care the authors have had in responding to my comments in a thoughtful way. One of main concerns about the paper was whether the truncation of the megafauna from the distribution of individuals is equivalent to comparing the distribution of abundances as function of body sizes across many different ecosystems (e.g. Figure 3). It is still not clear to me whether these two phenomena are theoretically equivalent, as the equilibrium distributions of the latter require long relaxation times and are often under evolutionary pressures. In contrast the truncation of megafauna may be followed by some short-term compensatory processes, something that the authors now discuss in much greater extent in SI1.

Figure 2 (unchanged in this submission) is still misleading in this regard. The downgrading of megabiota is represented both as the distribution curve ending at a lower body mass, but also having a number of individuals per unit area smaller. Without compensation, the number of individuals per unit area should remain the same for all classes below the truncation point, which is recognised by the authors in their rebuttal letter but which is not what is represented. With compensation, the number of individuals per unit area may increase for the classes below the truncation point, precisely the reverse of what is represented. I think the authors could still elaborate on these issues a bit further in the text and revise Figure 2 to make it more rigorous.

Some of the mathematical derivations have typos or are not clear. For instance, in S9 an integral side is missing before the last equality, and c_n in the last term is probably c_a . Below in the same page, $B_{tot}=M.A$, but I think it should be $B_{tot}=B.A$. Equation S_6 uses r_{max} and r_0 in the limits of the integral when it should be probably using m_{max} and m_{min} . Furthermore the reason to introduce S_6 here is not clear, as it does not seem essential for the rest of this section and then its derivation is repeated in S_11. Finally, I had already pointed out that I could not follow the derivation of S_13 into S_14, but the authors have not changed them. I still think S13 is incorrect as it is mathematically different from S14 (one scales with m_{max} other with $m_{max}^{5/4}$). I can understand how S14 is derived, but S13 seems to be the one for plants...

The other concerns I had with the paper in its previous version were mostly addressed by the authors. Overall, I believe these issues can be addressed by the authors and I think a revised version of the paper would be suitable for publication in Nature Communications and will inspire an interesting and exciting debate.

Reviewer #3 (Remarks to the Author):

Dear Authors, I really applaud your work and love the way you are addressing this interesting and important issue. The key functional role of soil microbes and terrestrial invertebrates is understated but still I like your work a lot and see it ready for publication. Even the rebuttal are sound and with good and strong arguments. All figures are excellent. Only try to avoid sentences like "the removal of megaherbivores in the Sahara would lead to almost an entire loss of all animal biomass/metabolism" because it is really not true (see for instance spiders and MTE in the Namib in the absence of any megafauna as in Mulder et al. (2019) *Acta Oecologica*) and you see it also clearly in the Future World of Fig. S8 (0% Change in the Sahara and almost no change in the Namib and the Kalahari). Few more things. Line 195, an autotrophic food web makes no sense, as any food web (and even a food chain) has an an autotrophic and heterotrophic componnt. Just write food web. Line 701, naturalist becomes Naturalist, Refs. 30, 35 and 37 are missing the page numbers, Line 786 *Balaenoptera musculus* in italics and Line 794 PLoS ONE. Again, my sincere compliments

Response to reviewers

Reviewers' comments:

Reviewer #2 (Remarks to the Author):

I have reviewed this manuscript in the previous round (Referee 5). I appreciate the care the authors have had in responding to my comments in a thoughtful way.

Thank you.

One of main concerns about the paper was whether the truncation of the megafauna from the distribution of individuals is equivalent to comparing the distribution of abundances as function of body sizes across many different ecosystems (e.g. Figure 3). It is still not clear to me whether these two phenomena are theoretically equivalent, as the equilibrium distributions of the latter require long relaxation times and are often under evolutionary pressures. In contrast the truncation of megafauna may be followed by some short-term compensatory processes, something that the authors now discuss in much greater extent in SI1.

Thank you.

Figure 2 (unchanged in this submission) is still misleading in this regard. The downgrading of megabiota is represented both as the distribution curve ending at a lower body mass, but also having a number of individuals per unit area smaller. Without compensation, the number of individuals per unit area should remain the same for all classes below the truncation point, which is recognised by the authors in their rebuttal letter but which is not what is represented. With compensation, the number of individuals per unit area may increase for the classes below the truncation point, precisely the reverse of what is represented. I think the authors could still elaborate on these issues a bit further in the text and revise Figure 2 to make it more rigorous.

We understand the reviewers point here but believe that this figure is likely still best. We tried a few alternative figures but thought they injected more confusion. For example, a version of Fig. 2 showing several size spectra functions overlapping (they are currently offset slightly) but differing in terms of the maximum size is not clear and would add more confusion. Overlapping functions that differ in their maximum size would be difficult to see and interpret (we played around with several variants and showed them to colleagues). Another option is to show several different size spectra functions that differ in their normalizations (their heights) but with similar slopes. However, this would also lead to confusion as the question then is why does the normalization change with size downgrading? Both we believe are also unrealistic. Differential mortality of the larger organisms (megabiota) will not just remove only the biggest organisms but also organisms that are close in size. As a result increasing chances of mortality of larger individuals will tend to steepen somewhat the distributions just as we have drawn in Fig. 2.. In an effort to move forward here, and hopefully address the concerns raised, we have revised the

text in the Figure legend of Fig. 2 to hopefully address this point.

Some of the mathematical derivations have typos or are not clear. For instance, in S9 an integral side is missing before the last equality, and c_n in the last term is probably c_a . Below in the same page, $B_{tot}=M.A$, but I think it should be $B_{tot}=B.A$. Equation S_6 uses r_{max} and r_0 in the limits of the integral when it should be probably using m_{max} and m_{min} .

Thank you. Yes, I caught those too and in the revision these have been corrected.

Furthermore the reason to introduce S_6 here is not clear, as it does not seem essential for the rest of this section and then its derivation is repeated in S_11.

We use equation S6 as a pedantic example to provide a general argument for how to scale up from the size distribution and individual allometry to a given ecosystem measure. As we have had questions previously on the derivation we think it is important to provide a general overview first then followed by specifics. The reviewer is correct that we repeat this method later but with specific ecosystem measures. We have clarified this section.

Finally, I had already pointed out that I could not follow the derivation of S_13 into S_14, but the authors have not changed them. I still think S13 is incorrect as it is mathematically different from S14 (one scales with m_{max} other with $m_{max}^{5/4}$). I can understand how S14 is derived, but S13 seems to be the one for plants...

Thank you for catching that. Eqn. S14 was indeed not correct as stated. Eqn. S14 is for predicting total animal biomass M_{Tot} (not the total resource utilization rate J_{tot} as was written). Eqn S13 is correct as stated. We have now streamlined the argument in the Sup. Doc. Eqn S14 is now not needed.

The other concerns I had with the paper in its previous version were mostly addressed by the authors.

Overall, I believe these issues can be addressed by the authors and I think a revised version of the paper would be suitable for publication in Nature Communications and will inspire an interesting and exciting debate.

We thank the reviewer for their careful reading, input, and excellent suggestions.

Reviewer #3 (Remarks to the Author):

Dear Authors, I really applaud your work and love the way you are addressing this interesting and important issue.

We thank the reviewer for their enthusiastic support

The key functional role of soil microbes and terrestrial invertebrates is understated

We agree and understand the reviewers point here. We have now tried to indicate that the focus of this paper is on large plants and animals but that future work should better assess the role of microbes and inverts.

but still I like your work a lot and see it ready for publication.

Thank you.

Even the rebuttal are sound and with good and strong arguments. All figures are excellent.

Thank you.

Only try to avoid sentences like "the removal of megaherbivores in the Sahara would lead to almost an entire loss of all animal biomass/metabolism" because it is really not true (see for instance spiders and MTE in the Namib in the absence of any megafauna as in Mulder et al. (2019) *Acta Oecologica*) and you see it also clearly in the Future World of Fig. S8 (0% Change in the Sahara and almost no change in the Namib and the Kalahari).

We have now modified this sentence.

Few more things. Line 195, an autotrophic food web makes no sense, as any food web (and even a food chain) has an an autotrophic and heterotrophic componnt. Just write food web.

Agreed.

Line 701, naturalist becomes Naturalist, Refs. 30, 35 and 37 are missing the page numbers, Line 786 *Balaenoptera musculus* in italics and Line 794 PLoS ONE.

Agreed. These changes have now been made.

Again, my sincere compliments

Thank you. We sincerely appreciate the time and effort in providing constructive reviews.